# Design and Long-Term Performance of a Pilot Wastewater Heat Recovery System in a Commercial Kitchen in the Tourism Sector

Jan Spriet, Ajeet Pratap Singh, Brian Considine *, Madhu K. Murali and Aonghus McNabola

Department of Civil, Structural & Environmental Engineering, Trinity College Dublin, D02 PN40 Dublin, Ireland; muralim@tcd.ie (M.K.M.); amcnabol@tcd.ie (A.M.)
* Correspondence: considib@tcd.ie; Tel.: +353-1-896-3837

**Abstract:** This paper assesses the performance of waste heat recovery from commercial kitchen wastewater in practice. A pilot study of heat recovery from the kitchen at Penrhyn Castle, a tourist attraction in North Wales (UK), is outlined. The pilot heat recovery site was designed and installed, comprising a heat exchanger, recirculation pumps, buffer tank and an extensive temperature/flow monitoring system for performance monitoring of the waste heat recovery system. Continuous monitoring was conducted for a period of 8 months, covering the 2022 tourist season. The recovered heat from the kitchen wastewater preheats the incoming cold freshwater supply and consequently reduces the amount of energy consumed for subsequent water heating. Retrofitting the pilot heat recovery system to the kitchen drains resulted in a heat saving of 240 kWh per month on average, a reduction of 928.8 kg $CO_2$e per year, and a payback period for the investment costs of approximately two years, depending on the cost of energy supply. The presented results illustrate the potential of this form of renewable heat in reducing the carbon footprint of water heating activities in buildings and the hospitality sector.

**Keywords:** wastewater heat recovery; heat exchange; renewable heat; heating load; energy saving

## 1. Introduction

The declaration of wastewater heat as a renewable energy source in the EU Green Deal has accelerated scientific research towards finding new technologies for wastewater heat recovery. These techniques aim to lower the carbon footprint of water-heating activities to meet net zero carbon emission targets [1]. Wastewater Heat Recovery (WWHR) reduces the total energy requirement for water heating and, subsequently, the associated heating expenses and carbon emissions [2]. This is achieved by the integration of heat exchangers, and sometimes heat pumps, within the wastewater drainage system to transfer waste heat from hot wastewater to other uses, such as preheating incoming cold feed water to a hot water system [3,4].

According to recent data, greater than two-thirds of the heat requirements for hot water production are derived from fossil fuels in Ireland and the UK [5]. The related greenhouse gases (GHG) released by the burning of conventional fossil fuels result in adverse climate change impacts, which raises concerns for human health and the environment [6,7]. Commercial and domestic buildings consume about 30% of global energy and emit 17% of GHG emissions. Around 20% of global energy consumption is associated with water heating activities [8]. Approximately 50% of the energy requirements in the commercial and domestic sectors are associated with water heating and steam generation to be used for various heating applications [9]. Water heating in domestic and commercial sectors has a large potential for further optimisation and needs effective management to reduce the overall building sector's heating and energy expenditure.

Considerable energy is utilised in water-heating applications within kitchens, hotels, cafes, restaurants, etc. Food and drink preparation consumes about 40–50% of the total energy consumption in commercial kitchens and a major portion of this energy is lost as waste heat-embedded in the wastewater discharge [10]. According to data estimated on wastewater heat recovery potential, the hospitality sector in the UK alone has a cumulative potential of 1.4 TWh/year [8]. Various technical approaches to WWHR have been proposed, suggesting different methods for thermal recapture from hot wastewater [4,8,9,11,12]. The waste heat recoverability depends on the type of system, point of recovery, wastewater flow rates and temperatures [13–16]. An experimental analysis to examine the thermal recapture potential from domestic kitchen dishwashers was conducted by Selimli et al. (2019), during which a finned tube heat exchanger was tested and shown to be capable of reducing energy use by 57.1 kWh [17]. A spiral coil heat exchanger has also been proposed for use in heat recovery in storage-based wastewater systems, with up to 60% of the embedded heat successfully recovered in some cases by this system [18]. Singh et al. (2023) proposed the integration of a heat exchanger in a grease trap for commercial kitchen WWHR applications and showed the potential to save up to 40% of the wastewater heat and also improve the retention of fat, oil and grease [11,12].

While there have been many methods proposed for wastewater heat recovery, significant research is still needed for the effective exploitation of this resource. In particular, previous research has focused extensively on the theoretical, desk and lab-based assessments of commercial kitchens [8,11,12] or, more broadly, on overall heat recovery from large hotels or resort drains [19]. Limited research has examined the design and performance of operational pilot-scale WWHR systems for commercial kitchens in the field. Furthermore, a WWHR system comprises more than just the performance of a heat exchanger. It may also require a buffer tank to store recovered heat for later use and, crucially, a method of integrating heat savings within the existing heating system of a building. The present research is novel in discussing WWHR in a real-world commercial operation, from the kitchen drain of the Tearooms at Penrhyn Castle, a popular landmark for tourists located near Bangor, UK.

This paper aims to present the findings from the operation of this pilot WWHR system, which was designed, installed, and operated for a complete tourist season in 2022. The performance of this system was monitored continuously through the measurement of relevant temperatures and flow rates. This paper details the performance and efficiency of the WWHR system based on these measurements alongside an analysis of its economic potential to offset fuel costs and environmental potential to reduce GHG emissions. The broader impact of these findings on WWHR from commercial kitchens was also considered, particularly in their design and to provide insights on potential operational issues.

## 2. Materials and Methods

### 2.1. Background

Penrhyn Castle is a historical tourist attraction receiving >200,000 visitors annually on average. The castle comprises extensive grounds, historic buildings and a café serving hot food and drinks with associated hot water consumption for food preparation and cleaning. Penrhyn Castle was chosen as an optimum location for heat recovery, considering it involved significant use of hot water in various activities. The hot water consumption during peak tourist season at the Castle is about 1000 m$^3$ on a monthly basis, which justifies the scope for WWHR investigations (See Figure A1).

The Tearooms, in particular, were selected for installation of the pilot plant following an extensive wastewater heat resource monitoring campaign at different locations within the Castle's sewer network (See Section 2.2). Figure 1 illustrates an aerial view of the Castle and some of the surrounding grounds, including the approximate location of the sewer network, heating system and pre-installation resource monitoring points.

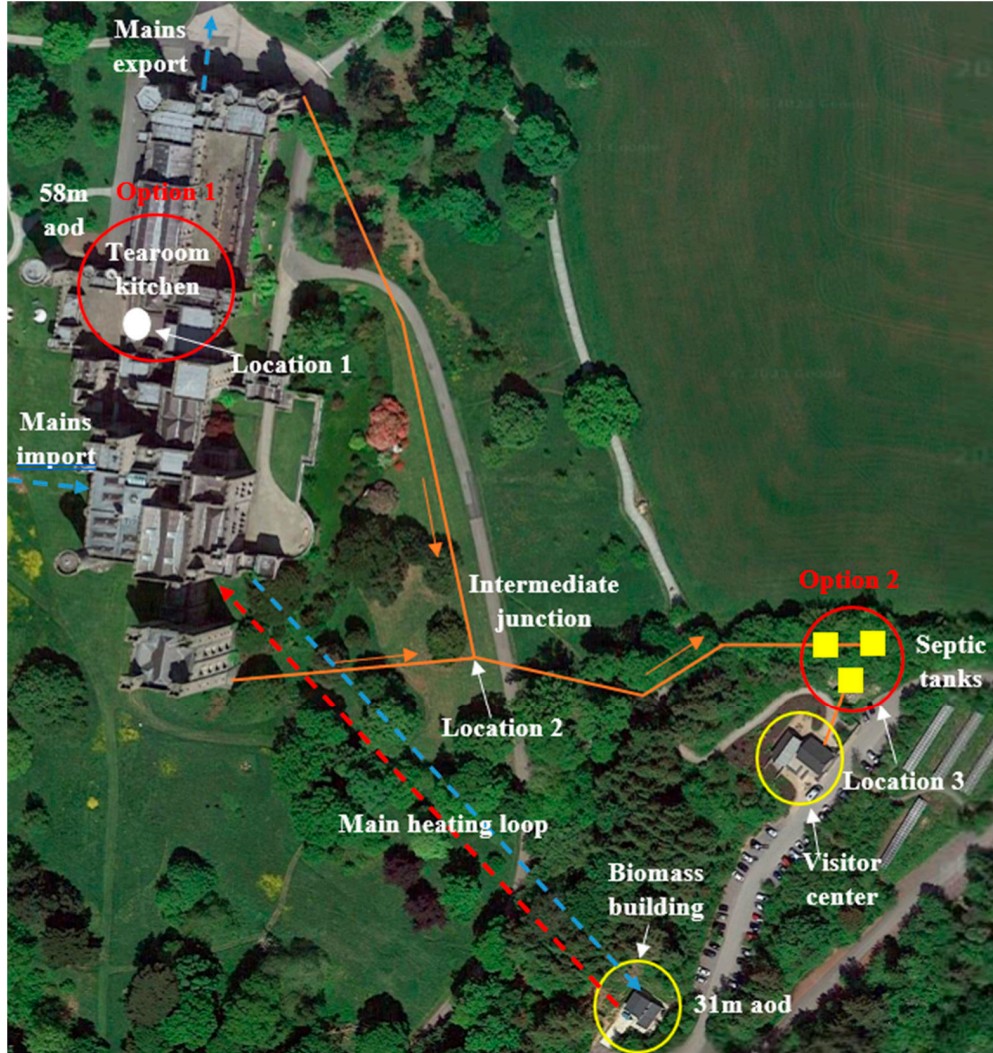

**Figure 1.** Measuring campaign at Penrhyn Castle for drain water heat recovery feasibility assessment.

Wastewater generation from the Tearooms kitchen is seasonal, with more tourists visiting the Castle in the summer period (see Figure A1). The Castle is closed during parts of the winter but has a year-round hot water demand from the on-site staff of the operator. The site was also chosen to help reduce the carbon emissions of a historic building and to improve the sustainability of heritage preservation in line with the site operator's objectives. Deep energy retrofits are often not possible in protected structures like this one, and thus, WWHR can play an important role in improving its carbon footprint. This is achieved through the capture and recycling of heat embedded in wastewater flushed down the drain and wasted to the environment prior to the installation of the pilot.

The current primary heating system is located in a biomass building (plant room), 250 m away from the Castle and at an elevation of about 30 m lower than the Castle (see Figure 1). Reducing heat consumption at the Castle would result in less biomass consumption and also reduced pumping costs for the main heating loop. The property is currently heated by two woodchip-based biomass boilers of 150 kW each (total 300 kW), represented as the traditional heating system in Figure A2 in Appendix A. The heat produced by the boilers is collected by a primary heating loop. The heat is then delivered to the secondary loop, also denoted as the main heating loop, by a plate heat exchanger. This is represented by the dashed lines in Figure 1 from the biomass building to the technical room local to the tearoom kitchen.

The following sections outline the details of the heat recovery system design and results obtained from the data monitored at the Penrhyn Castle Tearooms to show the thermal performance and payback period for operational and economic feasibility, respectively.

*2.2. Wastewater Heat Resources Monitoring Campaign*

The first stage in the design of a WWHR system was to assess the available waste heat resources and predict the feasibility (technical and economic) of installing the system at differing locations within a sewer system. Thus, a wastewater heat resources monitoring campaign was first carried out over several months in 2017 and 2018 to record wastewater flows and temperatures and assess technical viability in differing locations (e.g., distance between wastewater heat sources and the building's heating system).

The rural location of Penrhyn Castle means it relies on a system of septic tanks to manage its wastewater. These are located a distance from the main castle at a lower elevation (see Figure 1). In line with the findings of Nagpal et al. [16], the assessment of the wastewater heat resources at Penrhyn Castle was conducted to measure the available heat at the hot water application level, whole building level, sewer level, and within the wastewater treatment facility. In this case, the wastewater treatment facility was in the form of a series of large septic tanks. Therefore, as shown in Figure 1, monitoring of wastewater heat was conducted at the septic tanks, at the intermediate junction within the sewer network and the discharge point from the kitchen within the Castle. The heat resources were expected to have the highest flow volumes at the septic tanks, while the highest temperatures were expected to be located closer to the Castle. An assessment of which location contained the optimum balance of high energy and lower installation costs was the object of the measurement campaign. Table 1 illustrates the location, measurement type, period and frequency, and measurement devices used in the monitoring campaign. The Tinytag TG-4100 (Gemini Data Loggers (UK) Ltd., West Sussex, UK) records temperatures between $-40\ ^{\circ}\text{C}$ to $70\ ^{\circ}\text{C}$ with an accuracy range of $0.5$–$0.75\ ^{\circ}\text{C}$ above $0\ ^{\circ}\text{C}$ and was used at locations 1 and 3a–c. The hot and cold water flow rates at location 1 were taken from meters internal to the Penrhyn Castle existing systems. The temperature and flow rates for location 2 were measured using an ATEX MSFM sensor flow monitor (Dectronic Limited, Lancashire, UK) where the accuracy was $\pm 0.5\%$ and $\pm 2.5\%$, respectively.

**Table 1.** Summary of wastewater heat monitoring campaign [8,15,16].

| Location | Description | Measurement | Devices | Period | Frequency |
|---|---|---|---|---|---|
| 1 | Kitchen drain and water supply | Temperature and flow rate | Flow sensors and Tinytag TG-4100 | 19 June 2017–27 February 2018 | 1 min |
| 2 | Intermediate junction | Temperature and flow rate | ATEX MSFM sensor flow monitor | 27 February 2017–27 February 2018 | 15 min |
| 3a | Settlement tank-1 | Temperature | Tinytag TG-4100 | 20 March 2017–19 March 2018 | 5 min |
| 3b | Settlement tank-2 | Temperature | Tinytag TG-4100 | 20 March 2017–19 March 2018 | 5 min |
| 3c | Settlement tank-3 | Temperature | Tinytag TG-4100 | 20 March 2017–19 March 2018 | 5 min |

Initially, the location labelled as Option 2 in Figure 1 was selected for heat recovery resource monitoring. This location contained three settlement or septic tanks. These were Klargester-type tanks with two large (9000 L) units for the collection of wastewater from the Castle and one smaller unit (4500 L) for the collection of the wastewater from the visitor toilets, which can be seen as the small building adjacent to Option 2, in Figure 1. With sufficient heat resources and a WWHR system, incoming cold water to the Castle could be preheated here and sent to the biomass boiler building a short distance away (see Figure 1).

The distance between the WWHR source and the location where these heat savings can be integrated within the existing heating system is an important factor in the viability of WWHR systems. The longer this distance is, the more expensive it will be to lay pipework and perhaps provide pumping to connect the two systems. In addition, the longer this distance is, the greater the system losses will be where recovered heat is lost to the environment. The three septic tanks and biomass boilers were located at a similar elevation, which would avoid the need for additional pumping in this case, which was seen as an additional advantage to this location.

The tanks comprised an inner chamber for solid matter to settle and an outer chamber for wastewater accumulation. The wastewater temperature data collected showed a maximum of 13 °C during the 12-month sampling campaign, as illustrated in Figure A3 in Appendix A. This relatively low temperature meant that WWHR was unjustified at this location due to too high heat losses in the 250 m distance between the main Castle building and the septic tanks. In addition, evidence of groundwater infiltration within the sewer system at the mid-point acted to increase ambient heat losses further.

In parallel to this, monitoring was also conducted at the intermediate junction in the sewer network halfway between the septic tanks and Castle (see Figure 1). At this location, it was hoped that ambient heat losses could be reduced while wastewater volumes would be maintained. Unfortunately, negligible thermal differences persisted here due to ambient losses and groundwater infiltration. Thus, our monitoring campaign moved ahead towards Option 1, which is at the outlet of the Tearoom in the Castle.

The monitoring of drain water temperatures at the kitchen drain resulted in temperatures up to 58 °C, with median values ranging from 22–38 °C and an average daily flow of around 650 L (see Figure 2). The hot and cold water consumption within the kitchen was collected in order to account for drain water flow. This data was also used by Spriet et al. (2019) to generate synthetic load profiles for daily variations in wastewater flows and temperatures [8]. This was potentially an ideal location for heat recycling as the discharge point was in very close proximity to the heating interface unit (HIU) between the biomass boiler and the main castle heating loop (a 2–3 m distance). Therefore, recycled heat could be more easily integrated into the existing heating system at this point.

These conditions provided the opportunity for a direct heat recovery system whereby the waste heat is recovered using a heat exchanger only, and this heat was of sufficiently high temperature to avoid the need for a heat pump [16]. For these reasons, Option 1, at the kitchen drain, was selected as the pilot site. The temperature of the drain water is sufficiently elevated to directly preheat the mains water using the pilot heat recovery system (see Figure 3).

*2.3. Option 1 Pilot System Design*

Option 1, the direct heat recovery method, requires a heat exchanger, associated pipework components, and, in case the mains pressure is insufficient, a pump to circulate clean water through the heat exchanger. A direct buffer tank was also installed to store the preheated water, having a capacity of 300 L (see Figure 3a), to provide a consistent preheated water supply during the non-operational hours of the kitchen. Spriet et al. (2019) highlighted that a temporal mismatch exists between the generation of hot wastewater and the demands for hot water use and that WWHR system efficiency and economic viability could be improved in this regard by incorporating a buffer tank within the system to store saved heat [5]. The pilot system, in this case, incorporating a direct buffer tank, preheats the mains source water using the concentric shell and tube-type heat exchanger, and therefore, the required heat from the conventional heating system is reduced, resulting in a decrease in fuel consumption, related costs and emissions.

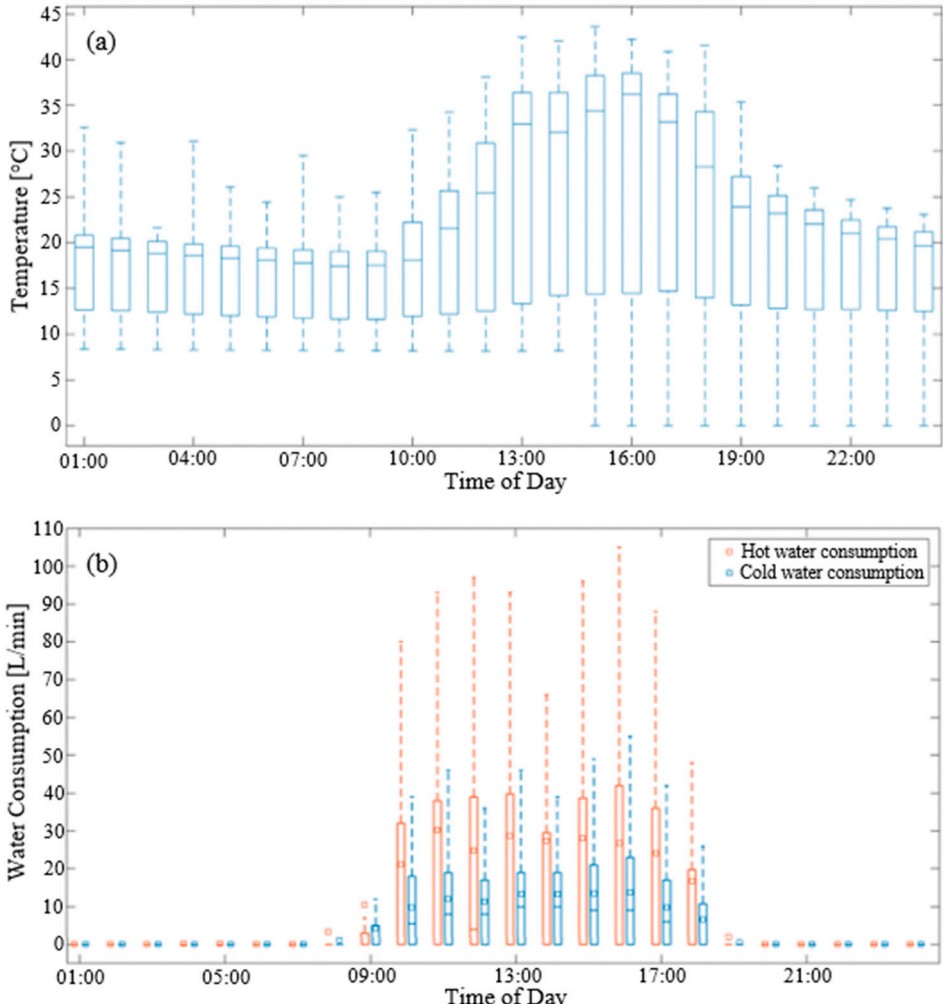

**Figure 2.** (**a**) Drain water temperature (**b**) Water consumption; monitoring at Option 1 recorded from June 2017 to January 2018.

The WWHR set-up was situated in a technical room where flow and temperature data were collected. This housed the buffer tank for storage of preheated water, an HIU, and a concentric shell and tube type high thermal conductivity copper drain pipe heat exchanger arrangement to achieve WWHR (see Figure 3). The two main locations of the WWHR system are: 1. Heat recovery from kitchen wastewater coming down the drain to preheat the cold freshwater (via a heat exchanger, see Figure 3b), and 2. HIU (see Figures 3c and A4), where hot water from the biomass boiler is indirectly exchanged with the preheated water from the WWHR system to supply freshwater at the desired temperature to the Tearooms.

The hydraulic installation of the WWHR system was first performed using copper components, and in a later stage, these were insulated against ambient heat losses using a lagging material (see Figure 3b). A commercially available concentric shell and tube counter flow heat exchanger was used to replace the kitchen drain, as shown in Figure 3b. The heat exchanger comprised a 168 cm long concentric pipe, 50 mm waste pipe diameter, with a capacity of 50 L/min of drain water and two 15 mm diameter freshwater connections. The wastewater flow could easily accommodate the flow rates shown in Figure 2b and was also matched to the existing 50 mm wastewater pipework. In the design stage, the effectiveness of the heat exchanger was estimated at 58%, taken as a weighted average from the manufacturer's freshwater flow rate data (See Appendix A Figure A4). This estimated effectiveness was taken as a weighted average of the expected effectiveness based on the manufacturer's information and the flow estimations in the drain of the kitchen at the tourist attraction in Wales. An interactive post-installation 3D view of the

pilot installation can also be viewed at https://dwruisce.github.io/PenrhynE/ (accessed on 1 September 2022).

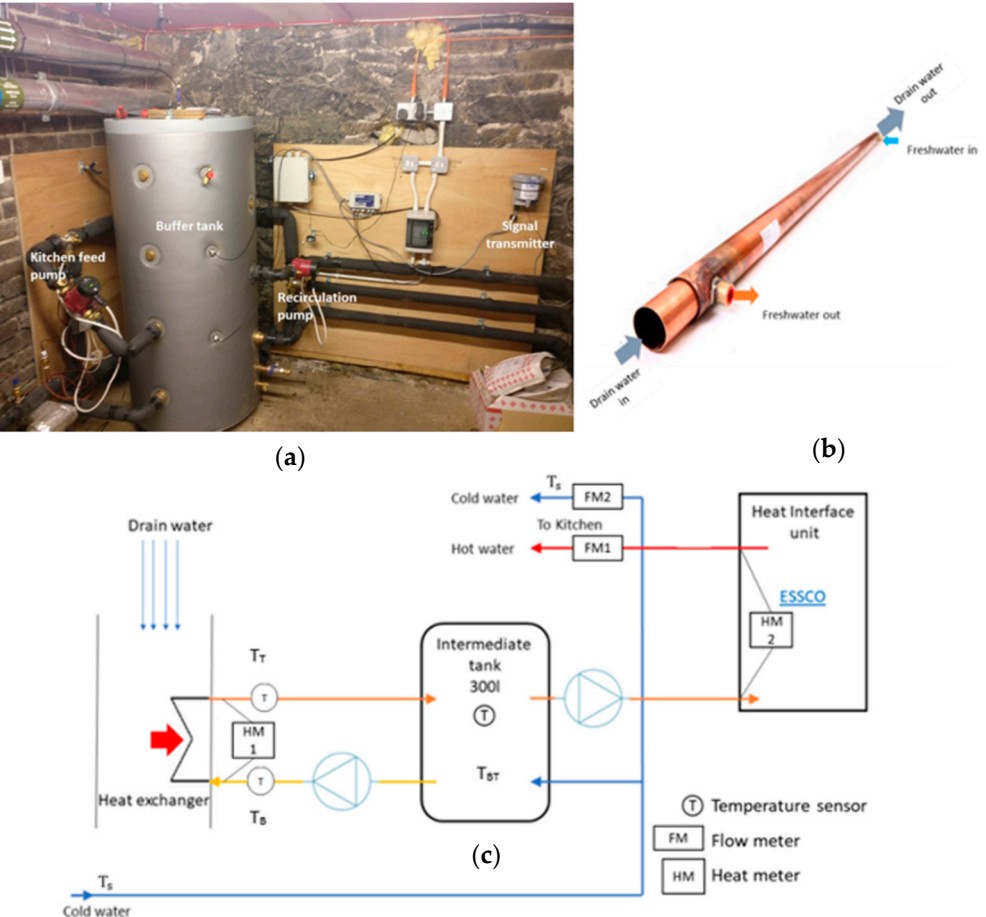

**Figure 3.** Pilot waste heat recovery system installation at Penrhyn Castle: (**a**) Actual pilot site image from technical room. (**b**) Shell and tube-type concentric HX used for heat recovery. (**c**) Schematic flow diagram of the overall WWHR system.

The energy supplied for water heating was derived from a woodchip-based biomass boiler. The pilot WWHR system was installed and placed in operation in Feb 2022 and operated for the full 2022 tourist season (February–September) to harness heat from the Tearooms at Penrhyn Castle. Continuous monitoring has been conducted to record temperature and freshwater flow at regular intervals at various locations of the installed unit to evaluate the amount of heat recovered, as shown in Figure 3c.

The logging and visualisation of data collected were provided by Detectronic and installed on-site after the installation of the hydraulic system. For data storage and monitoring, three transmitters were installed on site (see Figure 3a), collecting data in 10-minute intervals. The wastewater heat recovery system temperatures were monitored using PT-100 temperature sensors with an accuracy of $\pm0.30\,^{\circ}$C at 0 $^{\circ}$C (RS Pro, London, UK), which recorded temperature data at the top ($T_T$) and bottom ($T_B$) of the heat exchanger (HX) in the buffer tank ($T_{BT}$), in the cold water supply ($T_S$), and ambient air temperature ($T_a$). Freshwater circulation in the HX flow circuit was maintained via a circulation pump (Grundfos Alpha 2L). The freshwater flow rate from the HX was measured using a flow sensor with an accuracy of $\pm1\%$ (RS Pro Liquid Acetal Copolymer 4.5–16 V dc sensor, RS Pro, London, UK) and calibrated to provide 0.027 kg/s.

The system performance was assessed in terms of the magnitude of waste heat saved and recycled in the building's heating interface. The buffer tank accommodates preheated

water received from the heat exchanger and compensates for the time lag between hot wastewater generation and demand for hot water consumption. A time lag exists between when the heat is recovered and when it is required for supplementing hot water use. The direct buffer tanks enable the storage and optimisation of the recovered resources. Freshwater circulated continually through the concentric shell and tube heat exchanger ensures that the buffer tank always supplies preheated water during usage. A heating interface unit supplies additional heat from the main heating loop to further boost the preheated water from the buffer tank to the desired temperature. The HIU itself comprises another heat exchanger whereby heat recovered and stored in the BT is indirectly exchanged with heat provided by the Castle's main heating loop (provided by a biomass boiler). The incoming freshwater feed was preheated and stored in the buffer tank before it reached the heat interface unit. The HIU provides the interface which indirectly exchanges the thermal energy from the hot water generated by the biomass boilers and transported in the main heating loop with the preheated water from the WWHR system (see Figure A5 of Appendix A). The presence of the WWHR system results in a higher temperature for the freshwater going into the HIU. Therefore, the water requires less energy to reach a desired temperature of 60 °C.

In the case of the Tearooms at Penrhyn Castle, the distance between the HX at the kitchen drain, the buffer tank location, the HIU, and the existing heating loop is approximately 2–3 m. Buildings with larger distances between WWHR resource location and the location of the existing heating system would require additional capital investment not required here, which in some cases could make WWHR unviable. This paper investigates the case for economically viable recovery of wastewater heat using existing heat exchanger technology and new design configurations of the full system layout, considering all required system components. The paper examines whether the significant potential for energy and $CO_2$ emissions savings that exist for buildings with significant hot water usage can be exploited in practice.

### 2.4. Mathematical Expressions Used in Performance Analysis

The power needed to raise the temperature of cold freshwater from the supply temperature to the desired temperature level, i.e., generally 60 °C, was determined according to Equation (1). Equation (2) determines the power required in the HIU to heat freshwater to the desired temperature when the pilot heat recovery system is active/operational. No heat recovery took place during periods when no hot wastewater was discharged to the drain (e.g., at night-time when the Tearooms were closed or in winter periods).

The following mathematical expressions were used to determine the energy-saving performance of the system over time:

$$\text{Power requirement without heat recovery (watts)}, \ P_{w/o} = \dot{m} \, C_p \, (T_D - T_s) \tag{1}$$

$$\text{Power requirement (watts)}, \ P_{hr} = \dot{m} \, C_p \, (T_D - T_{BT}) \tag{2}$$

where $T_D$ is the desired hot tap water temperature, which is considered 60 °C, $T_s$ denotes freshwater supply temperature, $T_{BT}$ denotes preheated water stored in the buffer tank, $\dot{m}$ is the freshwater mass flow rate in the HX, and $C_p$ is the specific heat capacity of water. The power saved due to the installation of the pilot waste heat recovery system is given as:

$$\text{Power saving (watts)} = (P_{w/o} - P_{hr}) \tag{3}$$

Missing data during the monitoring campaign occurred on a number of occasions due to sensor failure at the $T_B$ location and due to challenges with remote connectivity. In these cases, the missing data was predicted according to the measured relationship between $T_B$, $T_T$ and $T_{BT}$. This relationship was found to follow Equation (4) with an $R^2$ adjusted of 87%

based on 16,866 simultaneous measurements of all three variables. Less than 10% of data recordings contained missing values:

$$T_B = 2.95 + 0.036 T_{BT} + 0.845 T_T \tag{4}$$

The Payback period ($P_b$) calculations were based on the cost of energy saved on an annual basis and are given as:

$$P_b = (Pilot\ WWHR\ system\ total\ cost) \Big/ (average\ annual\ cost\ of\ energy\ savings) \tag{5}$$

The pilot heat recovery system total cost was £1885, which comprised of an arrangement of concentric copper heat exchanger pipe (£495), a water pump (£364), a buffer tank (£400), labour cost (£525), and cost for additional piping materials (£100). The cost of the data monitoring and telemetry system is not included here as this would not be required in a commercial installation. Similarly, the cost of human resource time in the design of the system by the research team was also not included, and in a commercial system, this would be accounted for in a markup of the profit margin. As such, payback periods estimated here are based on the cost–price only and are an underestimate of the commercial cost.

The cost of energy saving was based on the rates of differing fuel types in the UK for the years 2021 and 2022, as shown in Table 2 [20,21]. While biomass was the actual fuel used on site, the impact of differing fuel types on payback was assessed as the provision of water heating using biomass in the hospitality sector is not a very common form of heating in the UK.

**Table 2.** Price of UK fuel costs used for payback period assessment.

| Fuel Type | 2021 | 2022 |
|---|---|---|
| Biomass | 0.042 £/kWh | 0.070 £/kWh |
| Electricity | 0.28 £/kWh | 0.34 £/kWh |
| Gas | 0.07 £/kWh | 0.103 £/kWh |

Greenhouse gas (GHG) emission savings calculations were based on the expression given as:

$$GHG\ savings = Power\ saved \times Emission\ rate\ corresponding\ to\ power\ generation\ of\ 1\ kWh \tag{6}$$

Additionally, GHG emissions are associated with the production of 1 kWh of electricity, considered as 0.28307 kg $CO_2$e/kWh [5].

$$Cost\ of\ energy\ saving\ per\ year\ for\ a\ specific\ water\ heating\ fuel = Energy\ saved\ per\ year\ in\ kWh \times cost\ of\ fuel\ required\ per\ kWh\ of\ power\ production \tag{7}$$

## 3. Results

### 3.1. Wastewater Generation

The number of visitors accessing Penrhyn Castle during the monitoring period is shown in Figure 4a. A strong relationship exists between the number of visitors and the generation of wastewater at the Tearooms. This illustrates that the resulting hot wastewater generation is dependent on tourism numbers and footfall in the Tearooms. As such, WWHR was expected to be higher during conventional holiday periods and not present all year round. Additionally, the time of the week is also expected to have a significant influence on WWHR, as there are a higher number of customers on average visiting Penrhyn Castle on the weekend, particularly on a Saturday.

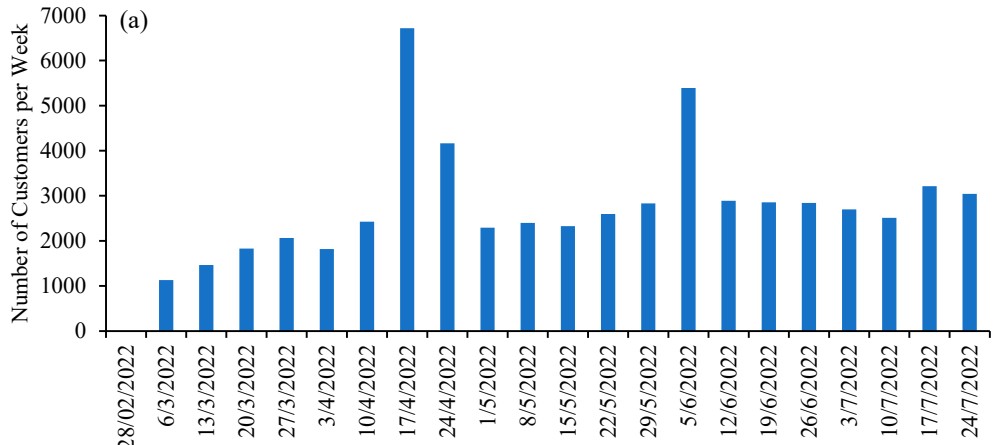

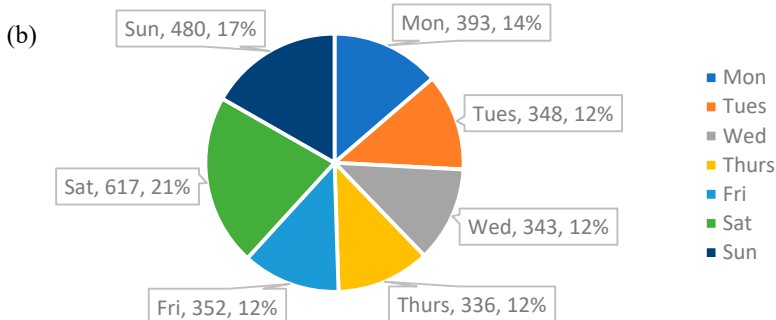

**Figure 4.** (**a**) Weekly variation in tourist visit numbers to Penrhyn Castle (**b**) Breakdown of the average daily visits within a week throughout the period 1 March 2022–24 July 2022.

### 3.2. Energy Savings

The freshwater supply at ambient temperature enters the vertical concentric shell and tube HX from the bottom and flows upwards opposite in the direction of the wastewater coming down the drain. A counterflow arrangement was held to attain the maximum level of thermal interaction between two heat-exchanging fluids (wastewater–freshwater). The maximum freshwater temperature after heat gain from the wastewater is 58 °C. However, the maximum thermal gradient between hot wastewater and cold freshwater is about 49 °C, while the minimum was about 18 °C during 2022. A high thermal gradient was mostly observed during kitchen operational hours, which is about 8 h a day when heat recovery is beneficial. The thermal gradient rapidly varies based on the type of food–drink preparation occurring in the kitchen facility, which is specifically based on customer choice and operational activity (i.e., dishwasher usage). On average, the preheated water temperature in the BT varies in the range of about 20 °C higher than the supplied freshwater temperature $T_S$ during kitchen operational hours (see Figure 5). This shows significant scope for saving thermal energy, which earlier was drained into sewers without recycling.

In Figure 5, the variation in freshwater supply temperature ($T_S$) and the temperature of the water after preheating by the HX ($T_T$) are shown. It was not possible to measure the temperature or flowrate of wastewater directly within the inner pipe of the HX as to do so would potentially risk the mixing of fresh and clean water in the HX, causing contamination risks, and because water in the inner waste pipe flows vertically down the pipe walls as a film flow, with a free surface. Aside from contamination risks, it is not possible to measure the temperature and flow rate of a film flow using minimally invasive conventional sensors. More invasive techniques to achieve this were not possible due to restrictions associated with the preservation of the historic structure at Penrhyn Castle.

Nonetheless, the amount of heat recovered from the wastewater was measurable from changes in the HX water temperatures and flow rates (Figure 6).

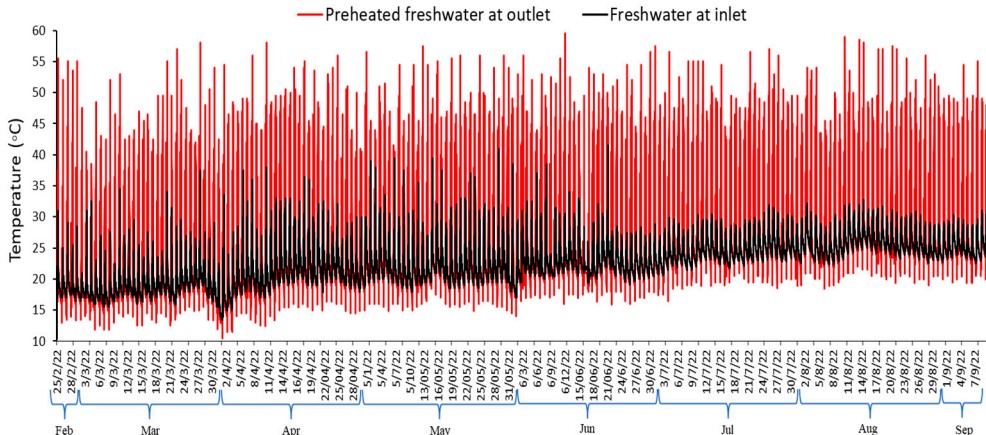

**Figure 5.** Monthly temperature variation of the freshwater from the supply temperature ($T_S$) and the preheated freshwater after heat recovery ($T_T$) from the hot kitchen wastewater.

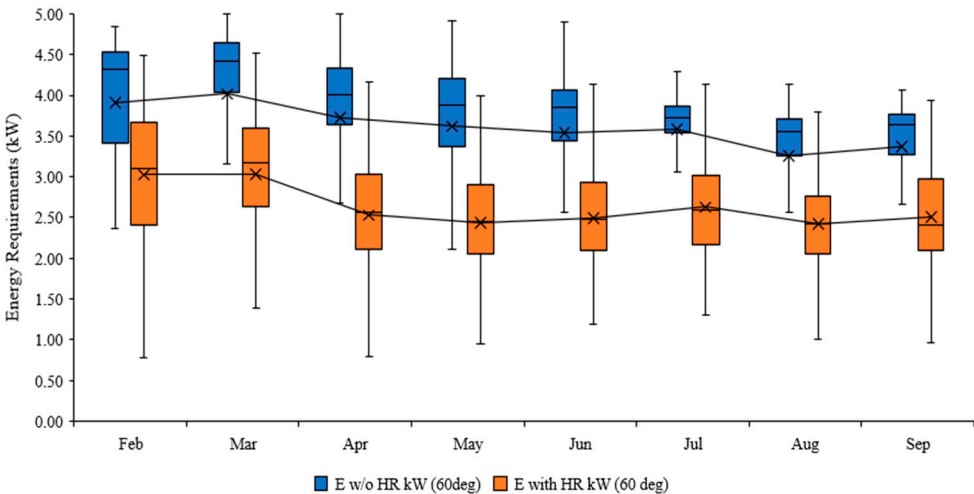

**Figure 6.** Comparative plot showing the average hourly water heating energy requirements with and without integration of the pilot heat recovery system at Penrhyn Castle.

Figure 6 displays the energy requirements for water heating with and without the integration of a pilot heat recovery arrangement for comparative analysis. The heat recovery calculated for the period February–September 2022 was 1770.9 kWh in total. The average monthly total heat recovery rate was observed to be about 240 kWh/month. However, the rate of heat recovery varies throughout the year and is highly dependent on the seasonal activity. The highest rates of heat recovery on average occurred in April, coinciding with high weekly customer attendance as illustrated in Figure 4 (associated with School holidays during Easter). WWHR results in a reduction in water heating energy requirements of 28% on average across the year.

Power savings from the heat recovery unit are shown in Figure 7, where a maximum of hourly average of approximately 2.49 kW was achieved during the summer period. The mean rate of hourly heat recovered ranged from 0.86–1.19 kW, and the median kW ranged from 0.73–1.28. There is more variability in the upper quartile range above the median values, especially for the maximum recovery rates, but below the median quartiles, the heat recovery minimum approaches zero. In all likelihood, this is linked to the kitchen activities and operation. Heat recovery is approaching zero as there is no wastewater generation

occurring at this point, and variability between the monthly maximum range is dependent on the level of customer activity. Furthermore, there is less variability when more heat recovery occurs, as seen with the shorter interquartile range for April and May, indicating more kitchen activity and a constant influx of customers.

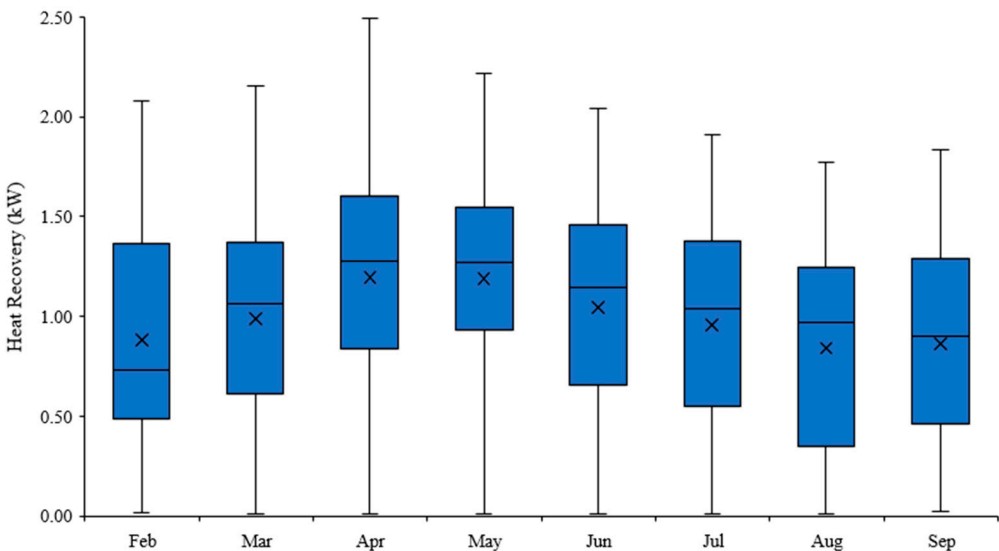

**Figure 7.** Average hourly heat recovery or power saving with the existing pilot heat recovery system at Penrhyn Castle in kW.

*3.3. Influence of Hourly, Time of Week and Visitor Numbers on Heat Recovery*

Figure 8 examines the influence of the daily visitor numbers on wastewater generation and subsequent daily heat recovery. Increased tourist attendance at Penrhyn Castle led to a greater rate of heat recovery on average, and the spread of data about the trendline is again attributed to the level of kitchen operational activities.

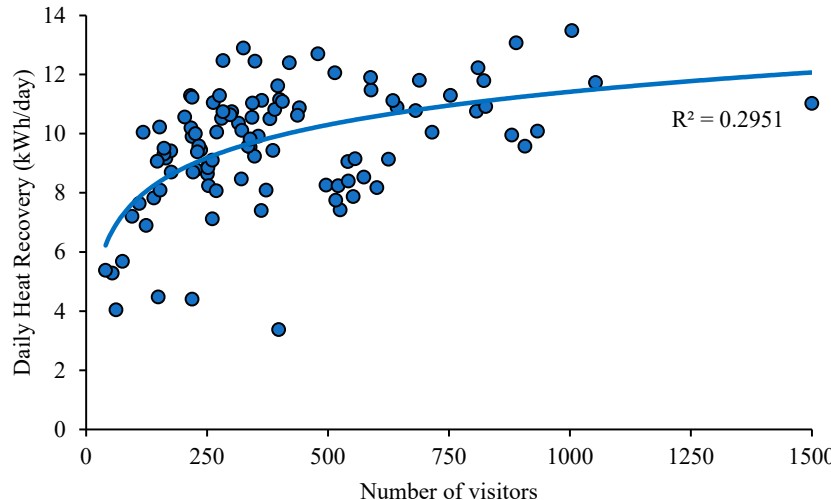

**Figure 8.** Influence of the number of visitors and the daily heat recovered from the existing pilot heat recovery system at Penrhyn Castle.

The rate of heat recovery was further disaggregated based on their operational hours and time of the week in Figure 9. The former shows a clear difference in heat recovery rate based on the time of the day. The Tearooms begins service at 10.30 am, and heat recovery either does not occur or is substantially lower between 9–11 am compared to other periods, particularly at lunchtime. Peak heat recovery rates occur immediately prior to, during and

after the lunch period. The Tearooms close to customers at 3.30 pm, and only post-closure, from 4 pm onwards, is a drop off in the rate of heat recovery from the wastewater observed. Kitchen staff will still be active after business closure in the 4–5 pm period, but all remaining heat in the wastewater system has been extracted after 5 pm.

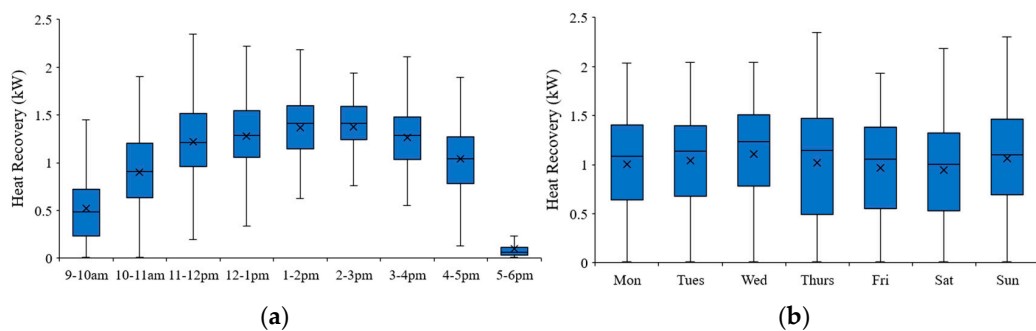

**Figure 9.** Variation in average hourly power recovery in kW for (**a**) operational hours and (**b**) time of the week.

The impact on the time of the week was also examined, and it was found that there were no significant differences in the rate of heat recovery. Considering Saturdays have nearly twice the average visitor rates than Thursdays, according to Figure 4b, kitchen activities remain steady throughout the week, and the monthly attendance is, more importantly, a driver of wastewater generation, as not all visitors to the castle also visit the Tearooms, and not all Tearooms visitor purchase food or drinks requiring hot water usage in preparation or cleaning.

*3.4. Ambient and Clean Water Temperature Impact on Heat Recovery*

Ambient temperature $T_a$ and the temperature of the clean water $T_S$ both can be expected to impact the thermal gradient available for heat recovery significantly. These two temperatures are also closely correlated with one another in the UK climate, i.e., both being colder in the winter and warmer in the summer. As such, colder periods will increase the thermal gradient between $T_a$ and $T_S$ which could be expected to increase the WWHR systems performance. However, colder periods would also increase losses to the ambient environment, noting that the system was well insulated to prevent this.

A typical variation of the rate of heat recovery under variable local ambient temperature conditions ($T_a$) during the kitchen operational hours is presented in Figure 10a, highlighting variations in WWHR due to varying levels of activity in the Tearoom (i.e., in line with morning opening times, lunchtime, and evening meal times). Lower freshwater temperature denotes a greater thermal gradient between upcoming cold water and hot kitchen wastewater. Since the pilot WWHR system was installed in close proximity to the kitchen wastewater outlet and is well insulated, this prevents major heat loss to the ambient environment, which improves energy-saving potential.

The cold freshwater temperature was found to be a little higher than the ambient temperature during monitoring, which lowers the scope for heat recovery due to a reduced thermal gradient. Figure 10a shows the relationship between the ambient temperature (Log $T_a$) and heat recovered, and Figure 10b relates the cold water supply temperature to the heat recovered. In both cases, the recycled heat rises as the ambient temperature and cold water supply increases, counter to the expected impact of the thermal gradients.

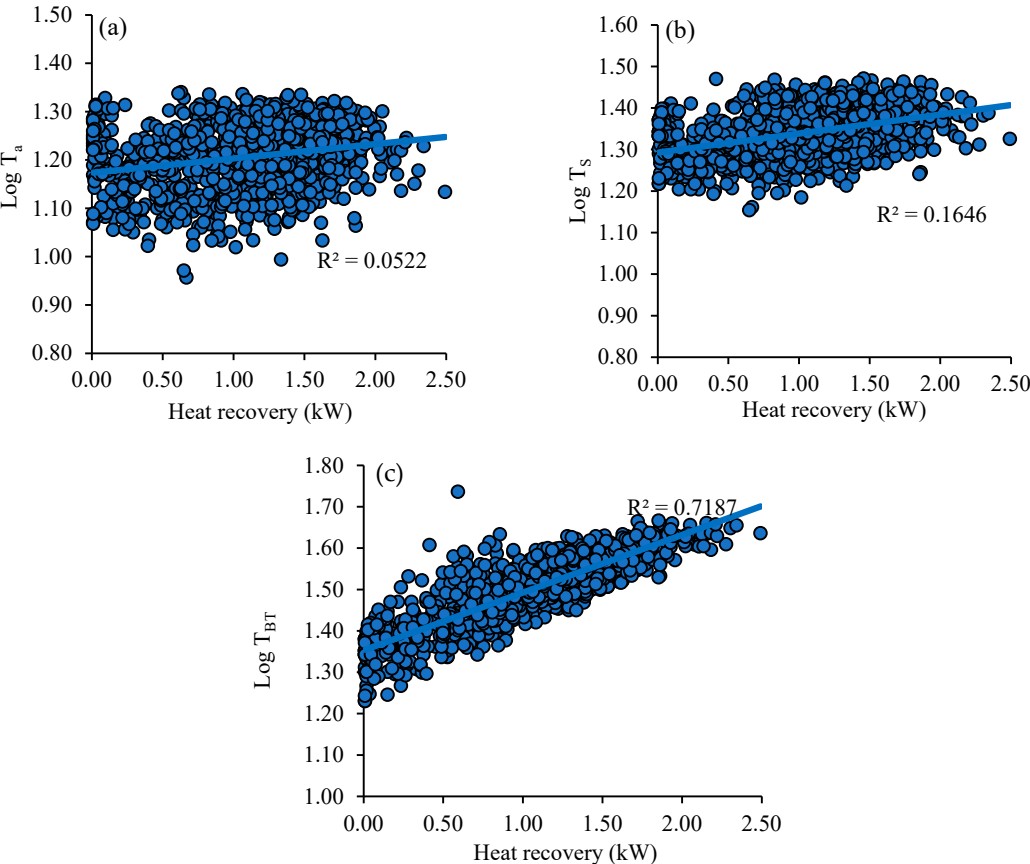

**Figure 10.** Relationship between the average hourly heat recovered in kW for (**a**) ambient temperature, (**b**) cold water supply, and (**c**) preheated water.

Examining the water temperature supplied to the buffer tank ($T_{BT}$) in Figure 10c after heat extraction alludes to a dominance of the wastewater temperature and kitchen activity. For example, if there was more kitchen activity due to higher tourist numbers and greater wastewater generation in the summer, higher thermal gradients will occur even with warmer cold water, compared to winter periods. Therefore, the expected theoretical relationship could not be ascertained from the field data and would require controlled experimentation with constant wastewater supply temperatures in order to assess the impact of the ambient and cold water temperature gradient simultaneously.

### 3.5. Economic Viability and Carbon Emission Assessment

The total cost of the present pilot heat recovery system at Penrhyn Castle was £1885. Figure 11 shows the estimated payback periods considering differing fuel types and their costs in 2021 and 2022. This was conducted to demonstrate the viability of WWHR installations of this nature over a wider range of hospitality venues where biomass is not present. In addition, the energy costs for biomass, electricity and gas for 2021 and 2022 were both used, considering that the 2022 prices were considerably higher than previous years and may not remain at this level in the long term.

Based on the amount of energy savings achieved, considering electricity as the fuel type used in water heating would result in a saving of approximately 919 £/year. This would result in a payback period for the system of 2.1 years. The associated reduced electricity consumption due to reduced primary water heating requirement would result in annual GHG savings of about 928.8 kg $CO_2$e/year. Electricity was the most expensive and carbon-intensive fuel type, and therefore, the installation of a WWHR similar to this one in a location using electric water heating stands to make the biggest economic and environmental impacts.

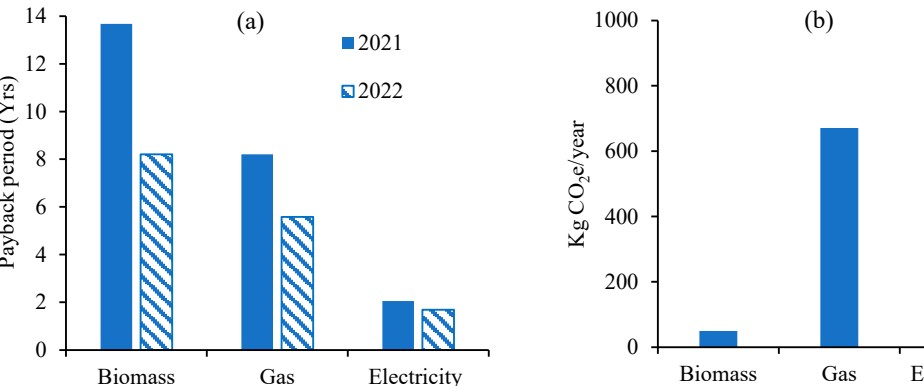

**Figure 11.** (**a**) Payback period (**b**) Annual Greenhouse gas emissions savings of the existing pilot heat recovery system at Penrhyn Castle for three major water heating fuels: biomass, gas, and electricity.

More broadly, considering the three fuel types, a payback period of 13.7, 8.2, and 2.1 years for biomass, gas, and electricity was estimated for 2021, respectively (Figure 11a). The value of energy savings for the different types of fuel amounts to 138, 230 and 919 £/year. The corresponding reduction in carbon emissions across the three fuel types was 49.4, 670.6, and 928.8 kg $CO_2$e/year for biomass, gas, and electricity, respectively (see Figure 11b). Fuel prices for the year 2022 were much higher compared to 2021 due to global inflation and geopolitical issues. In 2022, the payback periods were shortened to 8.2, 5.6, and 1.7 years for biomass, gas, and electricity, respectively. Finally, it should be noted again here that the cost estimates conducted excluded the cost of researchers' time and commercial profit margins.

## 4. Discussion

The wasted energy which was earlier discharged to the kitchen's drain has now been recovered by the pilot heat recovery system in the castle. Heat recycling attenuates the primary energy demand to be supplied from the traditional biomass boiler unit for heating the property. This reduces the wood pellet consumption, which results in monetary and greenhouse gas emissions savings. During the first year of operation, the savings were on course to achieve an 8.2-year payback. However, wood pellets for biomass boilers are a relatively inexpensive fuel type, and biomass boilers are far from the most common method of heating water in commercial kitchens. During 2022, gas and electricity fuel prices produced very attractive returns on investment for this system.

Nonetheless, while the system performance is strong, a number of factors limit the transferability of these findings to other commercial kitchen settings. These are connected with: 1. The heat exchanger design; 2. The proximity of the heating interface unit to the kitchen drain; 3. The climate; and 4. The nature of the tourism operation. A vertical concentric shell and tube counterflow heat exchanger was used here, which was commercially available and originally designed for use with showers. To achieve its maximum effectiveness of 58%, this HX must be oriented vertically to achieve a film flow and a minimum vertical height of 2 m was required below the outfall point from the kitchen for the installation. This was available at Penrhyn Castle but will not be available in all commercial kitchens.

Many commercial kitchen outfalls also contain a grease trap, which stores the hot wastewater for a period of typically 30 min to retain fat, oil and grease. This retention time would alter the temperature of the downstream discharge. In the case of Penrhyn Castle, a grease trap is not present as it is not required due to its historic building status. Therefore, in many commercial kitchen settings, an alternative HX design would be required, such as that proposed by Singh et al. (2023), which integrates the HX within the grease trap, possibly resulting in a differing HX effectiveness to the one achieved here [11,12].

As highlighted earlier, a key practical barrier to WWHR implementation is the challenge of what to do with the saved heat once it is recovered. There are a number of choices

for this, including using it to preheat the cold feed water to the existing heating system, as conducted here. However, as discussed earlier, in order for this integration to be viable, the distance between the waste heat source and the hot water heating system access point cannot be excessively long. Guidance on how far is too far is not available in literature as this is case-specific, depending on how much heat is available and how expensive it is to route the required piping between these two points. Future work could address the boundary of viability of this aspect.

The concentric shell and tube single-pass heat exchanger technology has been utilised primarily for WWHR in shower applications, but similar energy savings were found for commercial kitchens. For example, Wong et al. achieved energy savings of 4–15% by preheating the cold water supply for showers using a 50 mm diameter single-pass counterflow heat exchanger installed horizontally [4]. The Canadian Center for Housing Technology demonstrated a 9–27% energy savings in hot water generation for domestic showers using five vertically aligned heat exchangers [22]. In comparison, the WWHR applied to commercial kitchens also resulted in a 28% reduction in energy. Considering the WWHR for the shower system is local to the shower unit, the recovery potential is more limited when considering usage compared to the larger wastewater flows from a commercial kitchen and greater energy density available.

Aside from directly integrating the heat into the existing heating system, the saved heat could also be used as an input or part of the input to a heat pump for space or water heating. Wallin and Claesson highlighted that the low-grade heat savings from a WWHR system on a domestic scale could be used with a heat pump to reduce net heating demands by 33–34% using waste heat from a three-minute shower [23]. For that reason, where the aforementioned distance is too far, alternative options to use the saved heat can be found.

Finally, the results of this investigation are limited in their transferability to significantly different climates. WWHR in hotter climates has not received significant attention, possibly due to lower thermal gradients. The seasonal nature of the operation of the Tearooms within Penrhyn Castle also impacts the transferability of the results. Commercial kitchens with more intensive and full-time use are likely to show a stronger impact and return on investment than what we have found here for a relatively small café.

## 5. Conclusions

The present study focused on assessing the viability of heat recovery from commercial kitchen wastewater in practice to promote renewable energy to achieve net-zero carbon emission goals of the EU in the built environment. A pilot heat recovery system was installed to recover the waste heat from the kitchen wastewater released from the Tearooms at Penrhyn Castle, UK. The energy saving obtained via waste heat recovery results in reduced carbon emissions generated by the burning of fuels for water heating. The installed pilot heat recovery system was capable of preheating incoming cold water by 15 °C on average. The power saved aids in reducing heating-based expenses and $CO_2$ emissions. The system saves 1 kW of heating power on average during the 2022 tourist season, manifesting as 240 kWh per month and 928.8 kg $CO_2$e/year. The actual payback period for 2022 using biomass as the fuel source was 8.2 years, while this could be reduced to 5.6 or 1.7 years when gas or electricity was the fuel source. The results of this paper highlight that a significant and economically viable scope for WWHR exists in commercial kitchens in practice and that greater uptake and exploitation of this resource is warranted. Future research should focus on investigating the technology across the hospitality sector, targeting larger businesses such as hotels and entertainment venues. In addition, the technology should be trialled across different climate types and controlled experimentation to investigate the impact of the ambient temperature on the heat transfer process.

**Author Contributions:** Conceptualization, A.M. and J.S.; methodology, J.S., B.C., A.P.S., M.K.M. and A.M.; formal analysis, B.C. and A.M.; data curation, M.K.M. and A.P.S.; writing—original draft preparation, A.M. and A.P.S.; writing—review and editing, A.M. and B.C.; visualisation, A.P.S. and

B.C.; supervision, A.M.; project administration, A.M.; funding acquisition, A.M. All authors have read and agreed to the published version of the manuscript.

**Funding:** The authors would like to acknowledge that the work described in this paper has been partly funded by the European Regional Development Funds, Interreg Ireland-Wales Programme (2014–2020), through the Dwr Uisce Project, grant number 80910.

**Data Availability Statement:** Data available upon request.

**Acknowledgments:** The authors would like to acknowledge that the work described in this paper has been partly funded by the European Regional Development Funds, Interreg Ireland-Wales Programme (2014–2020), through the Dwr Uisce Project, grant number 80910. The authors would also like to acknowledge the support of National Trust Wales and Penrhyn Castle in facilitating the installation of the WWHR system. The views within this study are that of the authors and do not represent the funding agency or site operator.

**Conflicts of Interest:** The authors declare no conflict of interest.

**Appendix A**

(a)

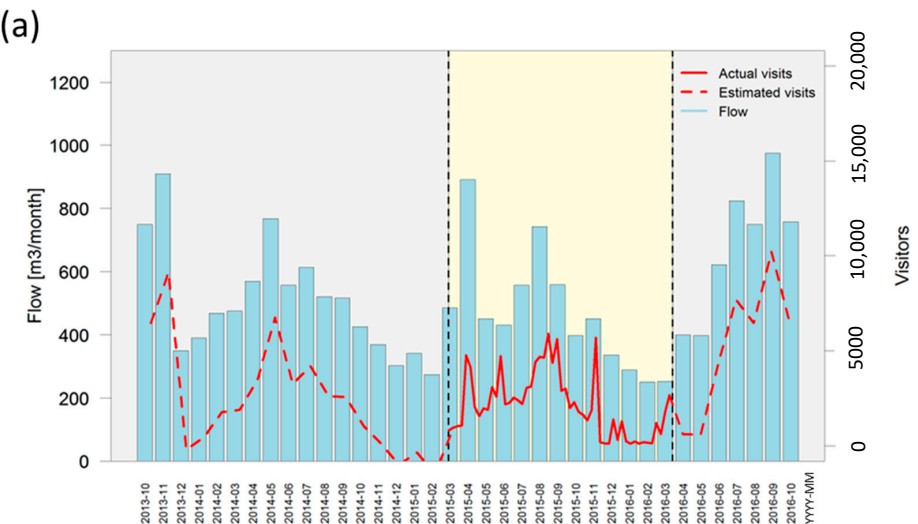

(b)

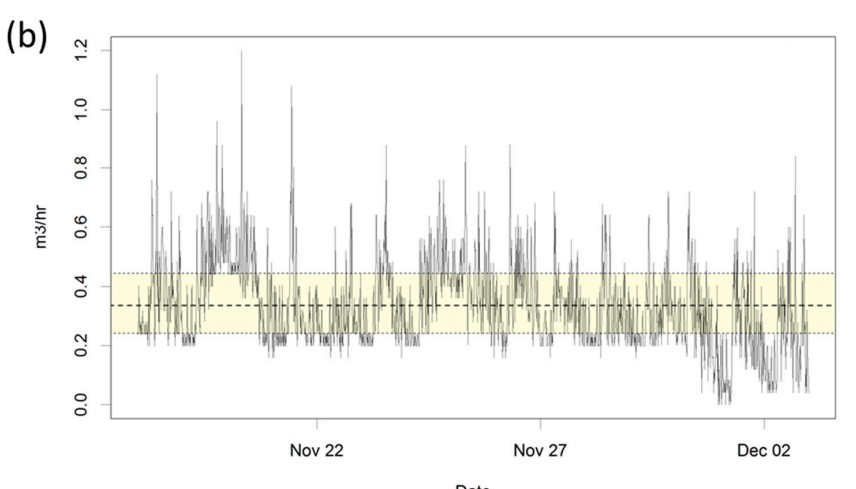

**Figure A1.** Visitor numbers and monthly water consumption at Penrhyn Castle. (**a**) monthly water flow rate and number of visitors, actual and estimated, during three consecutive years. (**b**) Flow meter data from 18 November 2009 to 3 December 2009. Note: $Q_{mean}$ = 0.33 m$^3$/h, $Q_{20}$ = 0.24 m$^3$/h and $Q_{70}$ = 0.44 m$^3$/h.

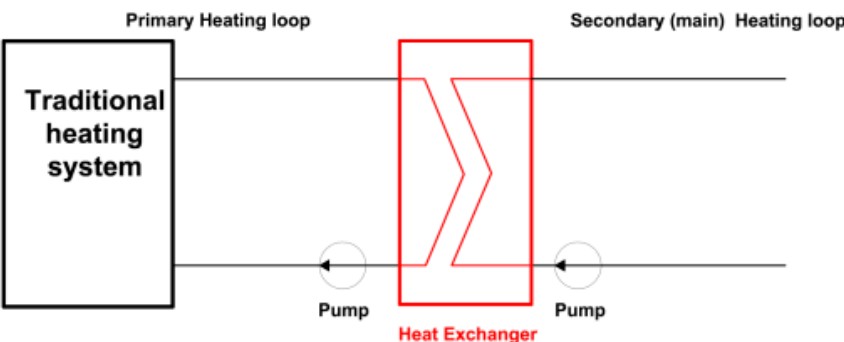

**Figure A2.** Primary and secondary heating loops.

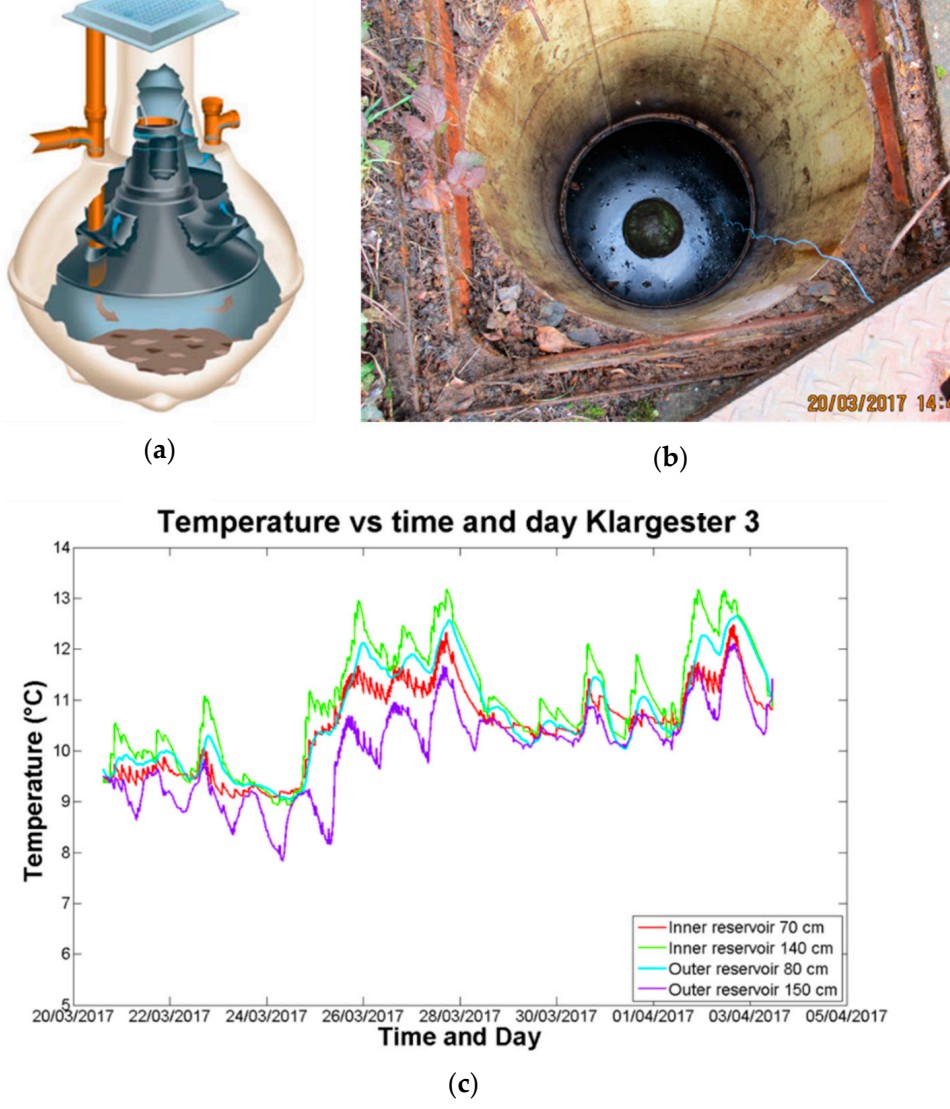

**Figure A3.** (**a**) Schematic of a Klargester settlement tank. (**b**) Overflow protection of the settlement tanks at Penrhyn Castle, Wales. (**c**) Temperature profile in the third settlement tank at Penrhyn Castle, Wales.

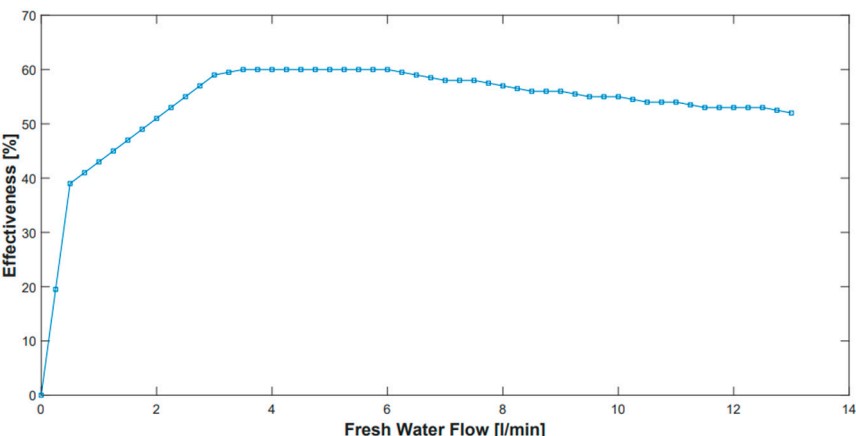

**Figure A4.** Effectiveness of heat exchanger based on the freshwater flow [24].

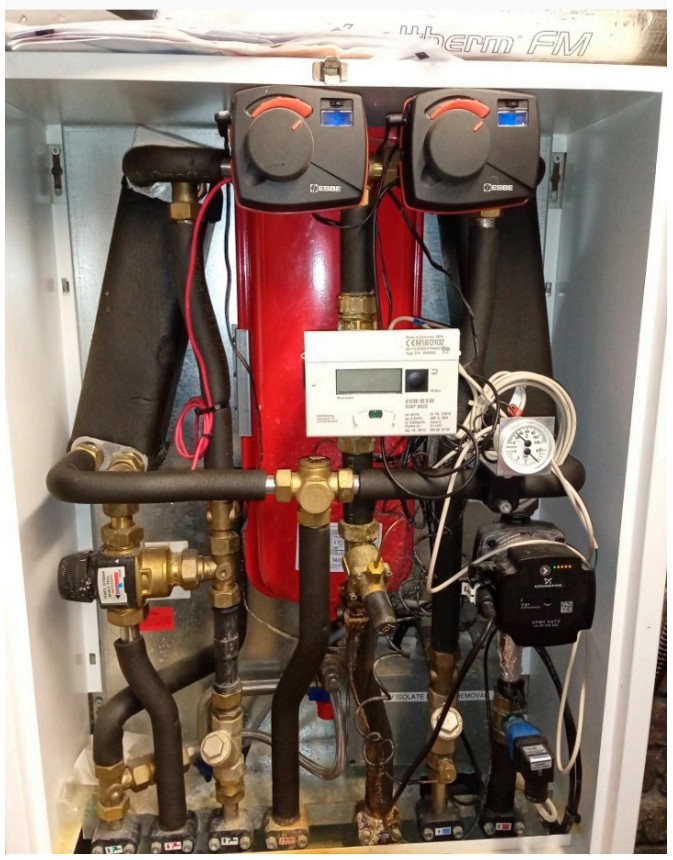

**Figure A5.** Heating interface unit of the pilot heat recovery system in the technical room.

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
