# Peer review of "Design and Long-Term Performance of a Pilot Wastewater Heat Recovery System in a Commercial Kitchen in the Tourism Sector"

_water, doi:10.3390/w15203646_

Round 1

Reviewer 1 Report

This study aims to present a study on the evaluation of a pilot wastewater heat recovery system application. It is in the scope of the journal. After reviewing the paper, I have the following observations and comments:

 1.     It is a topic of interest to researchers who are working on the waste water heat recovery (WWHR) applications. The knowledge in this paper can contribute to the applications of WWHR systems.

2.     The paper’s organization is not at an acceptable quality level. The material method is not rigorous/clear enough to correctly understand how the assessment was performed.

3.     There are a number of ambiguous points related to the presentation of the theory and the justification of the usage of the parameters. More information about the system components, concentric counter flow heat exchanger, pumps, tanks etc., should be useful to clearly understand the system. For example, what are the assumed inlet and outlet temperatures, the mass flow rates of drain water and heated water etc in the design stage? And, what is the real efficiencies in reality?

4.     Section 2.4 must be rearranged. Some data, notations, and presentations are not rigorous/clear enough to correctly understand the contents and their use. Since there are several flow lines (as drain water, hot water, etc.) in the system, the parameters, such as mass flow rates, must be represented with different subscripts to distinguish each of them.

5.     The equation numbers must be reordered, there are two Eq.5 at lines 279 and 298.

6.     The usage and the justification of the Eq.4 is not clear. How is it derived? Where is it used? It must be described more clearly.

7.     Comparisons and discussions with others’ works could improve the quality of the work.

8.     Units and their usage must be corresponded. Especially, how the power is converted to the energy?

9.      The quality of figures could be improved.

10. I suggest that a careful inspection throughout the paper must be performed.

I concluded that the present form of the paper must be revised according to the above comments. After that, further evaluation can be performed.

Reviewer 2 Report

Dear Authors,

Kind regards,

Reviewer

Author Response

see attach

Reviewer 3 Report

In the opinion of this reviewer, the manuscript should be ready for publication once the authors address the following minor comments:

The authors should point out the source of Eq. (4), reporting the significance of the fit using the adjusted R-squared and verify the assumption of normality of the residuals.

I would change Eqs. (1) - (3) and (5) by using symbols and defining them elsewhere. Keep it consistent.

Add the source [15,16] to Table 2.

The date time in Fig. 4(a) is cut off. Fix the issue.

The R-2 in Fig. 8 is very low (less than 50%). You should verify the hypothesis that this variable (number of visitors) significantly influences the daily heat recovery. What will happen if you consider other curve-fit equations?

The R-2 value should be reported on the plots in Fig. 10. The authors should also verify the normality of the residuals.

Reviewer 4 Report

This study evaluated the heat recovery in wastewater from

Commercial kitchens at Penrhyn Castle, a tourist attraction in North Wales (UK). The study is very interesting and important in the literature, however, before continuing with its processing, you must attend to the comments that are found in the attached file.

Round 2

Reviewer 1 Report

After reviewing the revised form of the paper, I have the following observation. The revised form of the paper is well organized, theoretical approach and experimental method are presented clearly. The subject is explained clearly, discussed, and compared properly with enough credit given to the contributions of the authors in this field.   Meaningful discussion and conclusions were made with the supporting data. The authors clearly responded to all the comments from the reviewers, and necessary revisions were performed accordingly. With the view of the above observations, I concluded that the present form of the paper could be acceptable for publication in the journal.

Reviewer 2 Report

Dear Authors,

The revised version of the manuscript is better than the original one. However, there are still some minor issues that should be addressed.

1) Your response to my question about the inlet and outlet temperature of the wastewater to/from the heat exchanger:

This data was not collected during the testing period in 2022, the temperature data from the wastewater was recorded from a manhole during an initial monitoring campaign from 2017-2018 (See Figure 2) for design purposes. For the WWHR system installation and testing in 2022, using a vertical HX pipe, it is not possible to measure the wastewater temperature within the HX. This operates with a film flow due to the partially filled nature of sewer pipe flow. As such a sensor cannot reliably measure this internal temperature, as the inside of the pipe contains mostly air. In addition, the test site location is a historic castle, and a protected structure. Modifications to the system to achieve additional monitoring were limited by the requirements to preserve the structure, and the limited vertical space available in the area in question.– I suggest putting this explanation in the manuscript, not just as an explanation to the reviewer.

2) In Figure 2a, the font size is smaller than in Figure 2b. I recommend enlarging the font in Fig. 2a.

3)  Revise the ‘Author Contributions’.

4) Figure A4 – if the data in Fig. A4 are from the manufacturer, you should add an appropriate reference.

Kind regards,

Reviewer

Reviewer 4 Report

The authors responded to my comments and it is now ready for processing.

Author Response

No further changes required for this comment. We thank Reviewer 4, for the positive assessment of the revised paper.